# Urinary and Serum Amino Acids May Be Associated with Podocyte, Proximal Tubule, and Renal Endothelial Injury in Early Diabetic Kidney Disease in Type 2 Diabetes Mellitus Patients

**DOI:** 10.3390/biomedicines13030675

**Published:** 2025-03-10

**Authors:** Maria Mogos, Oana Milas, Carmen Socaciu, Andreea Iulia Socaciu, Adrian Vlad, Florica Gadalean, Flaviu Bob, Octavian Marius Cretu, Anca Suteanu-Simulescu, Mihaela Glavan, Lavinia Balint, Silvia Ienciu, Iuliana-Lavinia Iancu, Dragos Catalin Jianu, Sorin Ursoniu, Ligia Petrica

**Affiliations:** 1Department of Internal Medicine II—Division of Nephrology, “Victor Babes” University of Medicine and Pharmacy, Eftimie Murgu Sq. No. 2, 300041 Timisoara, Romania; maria.mogos@umft.ro (M.M.); gadalean.florica@umft.ro (F.G.); bob.flaviu@umft.ro (F.B.); anca.simulescu@umft.ro (A.S.-S.); patruica.mihaela@umft.ro (M.G.); lavinia.balint@umft.ro (L.B.); ioana.ienciu@umft.ro (S.I.); iuliana.alioani@umft.ro (I.-L.I.); petrica.ligia@umft.ro (L.P.); 2County Emergency Hospital Timisoara, 300723 Timisoara, Romania; vlad.adrian@umft.ro (A.V.); jianu.dragos@umft.ro (D.C.J.); 3Centre for Molecular Research in Nephrology and Vascular Disease, Faculty of Medicine, “Victor Babes” University of Medicine and Pharmacy, Eftimie Murgu Sq. No. 2, 300041 Timisoara, Romania; carmen.socaciu@usamvcluj.ro (C.S.); sursoniu@umft.ro (S.U.); 4Research Center for Applied Biotechnology and Molecular Therapy BIODIATECH, SC Proplanta, Str. Trifoiului 12G, 400478 Cluj-Napoca, Romania; 5Department of Occupational Health, University of Medicine and Pharmacy “Iuliu Haţieganu”, Str. Victor Babes 8, 400347 Cluj-Napoca, Romania; andreea.socaciu@umfcluj.ro; 6Department of Internal Medicine II—Division of Diabetes and Metabolic Diseases, “Victor Babes” University of Medicine and Pharmacy, Eftimie Murgu Sq. No. 2, 300041 Timisoara, Romania; 7Department of Surgery I–Division of Surgical Semiology I, “Victor Babes” University of Medicine and Pharmacy, Eftimie Murgu Sq. No. 2, 300041 Timisoara, Romania; octavian.cretu@umft.ro; 8Emergency Clinical Municipal Hospital Timisoara, 300079 Timisoara, Romania; 9Department of Neurosciences—Division of Neurology, “Victor Babes” University of Medicine and Pharmacy, Eftimie Murgu Sq. No. 2, 300041 Timisoara, Romania; 10Centre for Cognitive Research in Neuropsychiatric Pathology (Neuropsy-Cog), Faculty of Medicine, “Victor Babes” University of Medicine and Pharmacy, Eftimie Murgu Sq. No. 2, 300041 Timisoara, Romania; 11Center for Translational Research and Systems Medicine, Faculty of Medicine, “Victor Babes” University of Medicine and Pharmacy, Eftimie Murgu Sq. No. 2, 300041 Timisoara, Romania; 12Department of Functional Sciences III, Division of Public Health and History of Medicine, “Victor Babes” University of Medicine and Pharmacy, Eftimie Murgu Sq. No. 2, 300041 Timisoara, Romania

**Keywords:** diabetic kidney disease, biomarkers, metabolites, endothelial dysfunction, proximal tubule damage, podocyte injury

## Abstract

**Background/Objectives**: The pathogenesis of diabetic kidney disease (DKD) is complex and multifactorial. Because of its complications and reduced number of diagnostic biomarkers, it is important to explore new biomarkers with possible roles in the early diagnosis of DKD. Our study aims to investigate the pattern of previously identified metabolites and their association with biomarkers of endothelial dysfunction, proximal tubule (PT) dysfunction, and podocyte injury. **Methods**: A total of 110 participants, comprising 20 healthy individuals and 90 patients divided in three groups were enrolled in the study: normoalbuminuria, microalbuminuria, and macroalbuminuria. Untargeted and targeted metabolomic methods were employed to assess urinary and serum biomarkers, as well as indicators of endothelial dysfunction, podocyte damage, and PT dysfunction through ELISA techniques. **Results**: Our research uncovered specific metabolites that exhibit varying levels across different sub-groups. Notably, glycine serves as a distinguishing factor between group C and the normoalbuminuric group. Furthermore, glycine is correlated with endothelial markers, especially VCAM. We observed a gradual decrease in kynurenic acid levels from group C to group P3; this biomarker also demonstrates an inverse relationship with both p-selectin and VCAM. Additionally, tryptophan levels decline progressively from group C to group P3, accompanied by a negative correlation with p-selectin and VCAM. Urinary tiglylglycine also differentiates among the patient groups, with concentrations decreasing as the condition worsens. It shows a strong positive correlation with nephrin, podocalyxin, KIM1, and NAG. **Conclusions**: In conclusion, glycine, tiglylglycine, kynurenic acid and tryptophan may be considered putative biomarkers for early diagnosis of DKD and T2DM progression.

## 1. Introduction

Diabetic kidney disease (DKD) is characterized by increased levels of albumin in the urine, decreased glomerular filtration rate (GFR), or a combination of both. It is a complication arising from type 2 diabetes mellitus (T2DM) and represents the leading cause of chronic kidney disease, potentially progressing to end-stage renal disease [1]. As a result of the rising population of individuals diagnosed with DM, there has been a significant increase in both the incidence and prevalence of DKD. The International Diabetes Federation estimates that this figure will grow from 10.5% in 2021 to 12.2% by the year 2045 [2,3]. As highlighted in numerous earlier research efforts, DKD is a multifaceted condition characterized by diverse pathogenic mechanisms [1,4,5]. Chronic hyperglycaemia is marked by persistent high blood sugar levels, which can cause harm to the glomerular filtration barrier. This damage extends to endothelial cells, the glomerular basement membrane, and podocytes. The filtration barrier is essential for filtering out substances such as water and solutes while retaining albumin. Deterioration of podocytes, alterations in the glomerular basement membrane, and mesangial expansion lead to the development of nodular sclerosis. This condition may also be linked with arterial hyalinosis and tubulointerstitial fibrosis in more advanced stages. Consequently, these changes result into the onset of microalbuminuria, which can eventually progress to macroalbuminuria [5].

As previously noted, the rise in DKD patients can be attributed to the absence of early indicators for the condition. In recent years, a developing collection of studies has emerged and emphasize serum and urine metabolites, along with initial biomarkers associated with podocytes, endothelium, and proximal tubules.

Our ongoing research is centred on the role of specific metabolites: tryptophan (TRP), kynurenic acid (KA), L-acetylcarnitine (LAC), glycine (Gly), tiglylglycine (Tyg), and taurine (Tau) at the levels of podocytes, tubules, and endothelial cells.

The aim of this study is to illustrate the potential connections between amino acids, identified through both untargeted ultra-high-performance liquid chromatography coupled with electrospray-ionization quadrupole time-of-flight mass spectrometry (UHPLC-QTOF-ESI+-MS) and targeted methods utilizing pure standards, and biomarkers indicative of podocyte injury, renal proximal tubule dysfunction, and endothelial dysfunction.

## 2. Materials and Methods

### 2.1. Patients and Compliance with Ethical Standards

This study was designed as a pilot and cross-sectional study, and it was conducted in the Nephrology Department and in the Diabetes and Metabolic Diseases Department, County Emergency Hospital, Timisoara, Romania from July 2021 to April 2022. After approval of the Ethics Committee for Scientific Research of “Victor Babes” University of Medicine and Pharmacy Timisoara (no. 28/02.09.2020) and the Ethics Committee for Scientific Research of “Pius Brinzeu” County Emergency Hospital Timisoara (no. 296/06.04.2022), 110 participants were included in this study, from which 20 were healthy control subjects and 90 were patients with long-standing T2DM. Based on UACR, we stratified the 90 patients with T2DM in three equal subgroups of 30 patients each: subgroup P1 (UACR < 30 mg/g), P2 (UACR: 30–300 mg/g), and P3 (UACR > 300 mg/g). As an inclusion criterion, we selected patients with T2DM, with a duration of minimum five years. In addition, hospitalized patients enrolled in the study were non-critically ill, and the samples were collected after initial biological investigations. Patients with T2DM with poor control of diabetes (HbA1c > 10%), active infections, neoplasia, glomerular disease, end-stage renal disease, and T1DM were excluded from the study. At the time of screening, all patients had received treatment with angiotensin-converting enzyme inhibitors (e.g., Perindopril) or angiotensin receptor blockers (e.g., Candesartan, Irbesarta) for at least 10 years, as well as oral antidiabetic agents, insulin, and/or statins. Written informed consent was obtained from all participants, and to maintain the confidentiality, we used numerical codes for each sample.

### 2.2. The Preparation of Samples

The blood samples were collected via venipuncture and transferred into sterile vacutainers that did not contain anticoagulants, while urine samples were placed in sterile vials. These specimens were preserved at −80 °C until analysis. For the preparation, 0.8 mL of a mixture consisting of pure HPLC-grade methanol and acetonitrile was added to each 0.2 mL of serum and urine separately. The mixtures were vortexed to precipitate proteins, subjected to ultrasonication, and then stored at −20 °C for 24 h to enhance protein precipitation. Following centrifugation, the supernatant was collected and filtered through nylon filters. It was then transferred into glass microvials before being loaded into the autosampler of the UHPLC system for injection. Furthermore, the supernatant was moved into an autosampler vial for HPLC-MS analysis. Quality control (QC) samples were also obtained and utilized as representative generic samples; these QC samples were injected at both the beginning and end of the experiment, as well as after every tenth injection during the analysis of study samples [6,7].

### 2.3. Analytical Methods

#### 2.3.1. UHPLC-QTOF-ESI+-MS Analysis

The metabolomic analysis was conducted utilizing ultra-high-performance liquid chromatography in combination with electrospray ionization–quadrupole-time of flight–mass spectrometry (UHPLC-QTOF-ESI+-MS). This was achieved using a ThermoFisher Scientific UHPLC Ultimate 3000 system, which featured a quaternary pump, a Dionex delivery system, and mass spectrometry detection equipment from MaXis Impact (Bruker Daltonics). Metabolites were separated on an Acclaim C18 column (5 μm, 2.1 × 100 mm, with a pore size of 30 nm from Thermo Scientific, Waltham, MA, USA) maintained at 28 °C. The mobile phase comprised 0.1% formic acid in water (component A) and 0.1% formic acid in acetonitrile (component B). The total elution time was established at 20 min, with flow rates set at 0.3 mL/min for serum samples and 0.8 mL/min for urine samples.

For serum samples, the gradient was as follows: from 90% to 85% A over the first three minutes, then from 85% to 50% A between three to six minutes, followed by a decrease from 50% to 30% A during the next two minutes (six to eight minutes), transitioning from 30% to 5% A from eight to twelve minutes, and finally ramping back up to 90% A by the end of twenty minutes. In contrast, the gradient for urine samples started similarly with a shift from 90% to 85% A over three minutes, then reduced from 85% to 30% A during the next three minutes, followed by a decline from 30% to10% A over two minutes. This was held isocratic until twelve minutes before increasing back up to 90% by twenty minutes.

An injection volume of five millilitres was utilized while maintaining a column temperature of 25 °C. Several quality control (QC) samples obtained from each group were analysed simultaneously for calibration purposes. An internal standard solution of Doxorubicin hydrochloride (*m/z* = 581.3209) at a concentration of 0.5 mg/mL was added alongside the QC samples.

The mass spectrometry parameters included a positive ionization mode (ESI+), calibration with sodium formate, a capillary voltage set at 3500 V, a nebulizing gas pressure maintained at 2.8 bar, a drying gas flow rate of 12 L/min, and a drying temperature regulated at 300 °C. The mass-to-charge ratio values targeted for separation ranged between 60 and 600 Daltons. Instrument control and data processing were managed via specialized software: TofControl version 3.2, HyStar version 3.2, Data Analysis version 4.2 (Bruker Daltonics, Billerica, MA, USA), alongside Chromeleon™ 6.8 Chromatography Data System (CDS) Software.

#### 2.3.2. ELISA Technique

The ELISA technique was utilized in order to assess the urinary biomarkers of endothelial dysfunction, such as V-CAM1 (Catalogue Nr. E-EL-H5587 Elabscience, sensitivity—0.94 ng/mL; detection range—1.56–100 ng/mL; coefficient of variance (CV) < 10%); and p-selectin (Catalogue Nr. E-EL-H6180 Elabscience, sensitivity—18.75 pg/mL; detection range—31.25–2000 pg/mL; coefficient of variance (CV) < 10%); biomarkers of podocyte injury, such as nephrine and podocalyxin (Catalogue Nr. E-EL-H2360 Elabscience, sensitivity—0.1 ng/mL; detection range—0.16–10 ng/mL; coefficient of variance (CV) < 10%); and of proximal tubule dysfunction, such as KIM-1 (Catalogue Nr. E-EL-H6029 Elabscience; sensitivity—4.69 pg/mL; detection range—7.81–500 pg/mL; CV < 10%) and NAG (Catalogue Nr. E-EL-H0898 Elabscience; sensitivity—0.94 ng/mL; detection range—1.56–100 ng/mL; CV < 10%).

### 2.4. The Integration of the Results and Statistical Analysis

The present study builds upon earlier research by focusing on the identification and correlation of specific metabolites with markers of dysfunction in endothelial cells, proximal tubules, and podocytes. To achieve these findings, our initial review [6] involved collecting samples from patients diagnosed with diabetic kidney disease (DKD) and conducting statistical analyses utilizing both untargeted multivariate and univariate metabolomic methods. This analysis included a multivariate comparison between controls (group C) and all patients (the overall group P) based on the final data matrices. The differentiation between these two groups was illustrated through Fold Change, Volcano test results, Partial Least Squares Discriminant Analysis (PLSDA), and Variable Importance in Projection (VIP) values, which incorporated cross-validation parameters. Subsequently, we employed a Random Forest-based prediction test to compute *p*-values. Receiver Operating Characteristic (ROC) curves along with area under the curve values were derived from biomarker analysis. In the second phase, univariate analysis facilitated comparisons among subgroups P1–P3 against group C using one-way ANOVA, PLSDA, Random Forest techniques, and Heatmaps. Consequently, several potential biomarkers indicative of differentiation were identified.

In our follow-up investigation [7], we focused on 5 to 10 distinct molecules found in serum or urine and utilized the intensity of mass spectrometry peaks to compare various patient groups, such as between groups C and P, or among subgroups P1 to P3. For quantitative assessment, we identified particular biomarkers and constructed calibration curves using pure standards.

All the information outlined previously enabled us to identify metabolites that may be valuable for the early detection of DKD, including glycine, its derivative tiglylglycine, kynurenic acid, and tryptophan. Furthermore, we examined their relationships with proximal tubular, podocyte, and endothelial markers.

## 3. Results

### 3.1. Clinical Features and Biological Results

Table 1 presents the data resulting from history and the clinical examination, the common biological parameters, and those resulting from the ELISA assay and UHPLC-QTOF-ESI+-MS techniques. The data are presented as means ± standard deviations (SDs). The comparison between group C and patients was realized using the one-way ANOVA with Bonferroni correction, a chi-squared test, and a Kruskal–Wallis test, whereas the comparison between subgroups, namely C vs. P1, P1 vs. P2 and P2 vs. P3, was carried out by applying Student’s *t*-test, the chi-squared test, and a Mann–Whitney test.

### 3.2. Correlation of Serum Metabolites with Markers of Endothelial Damage

#### 3.2.1. Univariable Linear Regression Analysis

First, we employed the univariable linear regression analysis to identify metabolites associated with DKD, particularly those that distinguish between subgroups. The results indicated that serum glycine, serum KA, and serum TRP exhibited negative correlations with all markers related to proximal tubules, endothelial cells, and podocytes. Furthermore, we found a negative correlation between urinary glycine and urinary tiglylglycine, while a positive correlation was observed between urinary TRP and the aforementioned markers.

#### 3.2.2. Multivariable Linear Regression Analysis

The subsequent phase in identifying predictive models for metabolites involves multivariable linear regression analysis, which demonstrates a strong correlation between Tiglylglycine and P-selectin, podocalixin, and KIM-1.

### 3.3. The Impact of Tubular, Endothelial, and Podocyte Damage Biomarkers in DKD

#### 3.3.1. Kidney Injury Molecule-1

Kidney injury molecule-1 (KIM-1) is characterized as a type I transmembrane protein that is absent in healthy kidneys but emerges in the renal proximal tubule following injury. Van Timmeren MM et al. outline various mechanisms through which the buildup of proteins results in tubular blockage and mechanical strain, which subsequently activates tubular cells and stimulates the synthesis of KIM-1. Additionally, another mechanism is noted in diabetic patients, where tubular cells are activated by a harmful infiltrate, leading to the production of pro-inflammatory factors and dysfunction in the tubulointerstitial segment [8].

Our findings indicate increased levels of KIM1 among the different patient groups, particularly within the normoalbuminuric cohort, a conclusion that aligns with previous research concerning early DKD [9,10,11], data that can be found in Table 1. 

#### 3.3.2. N-Acetyl-β-D-glucosaminidase

N-acetyl-β-D-glucosaminidase (NAG) is an enzyme found in lysosomes, which is primarily located in the proximal tubules of the kidneys and excreted in minimal amounts through urine [12]. Injury to the epithelial cells within the proximal tubule leads to an excessive buildup of NAG. Furthermore, since NAG plays a role in carbohydrate metabolism, it is suggested that higher concentrations of NAG result from the exposure of proximal tubular cells to increased glucose levels [13].

Our study evaluated and found significantly higher levels of NAG in patients categorized as group P3 when compared to group P1, with notable statistical significance in P1 (see Table 1).

The findings were similarly reported by Quinghua Huang et al., who noted that the urinary NAG-to-creatinine ratio serves as a significant predictor, as well as by Mohammadi Karakani et al., regarding the assessment of urinary enzymes. Additionally, Miyachi E. et al. focused on the activity of urinary angiotensin-converting enzyme in patients with T2DM, while Skrha J et al., conducted a six-year follow-up study examining the correlation between NAG levels and albuminuria [14,15,16,17]. Collectively, these studies highlighted that levels of urine NAG rise with the progression of nephropathy, demonstrating a significant distinction between early and advanced DKD, particularly showing elevated NAG levels in patients exhibiting macroalbuminuria.

#### 3.3.3. Nephrin

Nephrin is a protein that spans across membranes and plays a crucial role in the area between the foot processes of podocytes in the kidney. In cases of hyperglycaemia-affecting podocytes, there is an alteration in nephrin regulation, resulting in nephrinuria, particularly observed in patients with normal albumin levels [18].

We observed elevated nephrin levels in patients with diabetes when compared to healthy individuals (data presented in Table 1). This particular biomarker has received limited research attention, but we identified several studies that address its role in diabetic mice versus their healthy counterparts. Of interest, several references include Kandasamy’s work on nephrin as an early indicator of glomerular injury, as well as studies by Chang et al., focusing on DKD in FVB/NJ Akita mice. Additionally, Ganesh Veluri et al. and Irena Kostovska et al., highlighted that urinary nephrin serves as a more sensitive and specific marker for diabetic nephropathy than microalbuminuria, finding significant nephrin levels in both groups of individuals with T2DM compared to healthy controls [18,19,20,21].

#### 3.3.4. P-Selectin

Patients suffering from type 2 diabetes mellitus experience systemic inflammation that may result in kidney damage. This inflammatory response causes an influx of monocytes and lymphocytes into the renal tissue, which in turn release pro-inflammatory cytokines, chemokines, and adhesion molecules such as VCAM and P-selectin [22].

Our research on p-selectin indicated a gradual rise in urinary levels of this biomarker, progressing from the P1 group to the P3 group. Similar patterns for this marker were observed in studies by Feng Wang et al., which examined the clinical significance of plasma CD146 and p-selectin, as well as by Khalid al Rubeean et al., who investigated IL-18, VCAM, and P-selectin as early biomarkers [22,23].

#### 3.3.5. Podocalixin

The upper surface of glomerular podocytes comprises a transmembrane protein known as podocalixin (PCX). In cases where podocyte function is impaired, this protein can leak into the urine. Consequently, elevated concentrations of PCX have been detected in the urine of individuals with diabetes [24].

Podocalyxin demonstrated statistical significance in our analysis, particularly when comparing the P1 group to the C group and within the subgroups P2-P3, consistent with findings from other studies. Hui Xie et al. noted that PCX is a positive marker, Hara M. et al., identified it as an early indicator, Vestra M. et al., linked it to podocyte injury in diabetic nephropathy, and Weil E. et al., discussed its role in podocyte detachment and reduced glomerular capillary function. These studies indicated a correlation between urinary PCX levels and UACR levels, as well as elevated urinary PCX levels in patients who exhibited normal urinary albumin values [25,26,27,28].

#### 3.3.6. VCAM1

VCAM1 is yet another molecule that plays a role in the adhesion of leukocytes to endothelial cells, facilitating the gathering of leukocytes during inflammatory responses. In summary, there is an elevated expression of VCAM in the kidneys of individuals diagnosed with DN [29]. Numerous studies indicate that VCAM is linked to the progression of diabetic kidney disease in individuals with diabetes, aligning with our findings [30,31].

### 3.4. Interpretation of Data Subsequent to Statistical Evaluation

Serum Glycine

In Table 1, it is evident that glycine differentiates between group C and the normoalbuminuric group, as well as between the P1 group and the P2-P3 groups, with levels declining as diabetic kidney disease advances. Additionally, it can be noted that glycine shows a correlation with endothelial markers, particularly VCAM, achieving a statistically significant value (*p* < 0.001) as it is shown in Table 2. 

Serum TRP and kinurenic acid

Our findings indicated a gradual decline in tryptophan levels from group C to group P3, along with a negative correlation observed with p-selectin and VCAM. Additionally, we noted a clear distinction between healthy individuals and those in the normoalbuminuric group.

Also, in our article, we observed a gradual reduction in S KA from group C to group P3, with a clear distinction noted between group C and normoalbuminuric patients. Additionally, in Table 2 is highlighted that this biomarker exhibits an inverse correlation with p-selectin and VCAM, as evidenced by highly significant *p*-values (*p* < 0.001 for p-selectin and *p* < 0.05 for VCAM).

Urinary Glycine

In urine, glycine demonstrated significant variations among the subgroups from group C to group P3. It demonstrates strong negative correlation with nephrin, while showing a weaker correlation with podocalyxin and N-acetyl-β-D-glucosaminidase (NAG).

In our examination, we observe noteworthy levels of this metabolite that effectively distinguish between different patient groups, with diminished concentrations noted as the condition progresses. Moreover, this metabolite shows a strong positive correlation with various markers of podocytes and proximal tubules, including nephrin, podocalyxin, KIM1, and NAG.

Urinary TRP

TRP is a metabolite that has been the subject of extensive research. In urine, TRP effectively distinguishes group C from group P1, as well as group P2 with group P3, due to a notable reduction in concentration levels observed in normoalbuminuric group, achieving statistical significance. It is important to note that group P1 indicates markedly low values, which subsequently increase threefold in groups P2 and P3 in comparison to group P1. Additionally, we identified a strong and positive relationship involving podocalyxin, KIM1, and NAG.

Urinary Tiglylglycine

In our examination, we observe noteworthy levels of this metabolite that effectively distinguish between different patient groups, with diminished concentrations noted as the condition progresses. Moreover, this metabolite shows a strong positive correlation with various markers of podocytes and proximal tubules, including nephrin, podocalyxin, KIM1, and NAG, as it can be seen in Table 3.

## 4. Discussion

In our study the presence of glycine, kynurenic acid, and tryptophan in serum indicates their potential role as biomarkers for renal endothelial dysfunction. Moreover, in urine, all metabolites—including glycine, tiglylglycine, and tryptophan—correlated with indicators of podocyte injury and proximal tubular dysfunction.

### 4.1. Serum Indicators of Endothelial Injury in Initial Stages of DKD

#### 4.1.1. Serum Glycine May Be a Marker of Endothelial Dysfunction in Early DKD

Glycine, categorized as a “nonessential” amino acid that the body can produce from serine, has been consistently linked to a negative correlation with T2DM. Research indicates that individuals who are nondiabetic but exhibit insulin resistance or impaired glucose tolerance show decreased levels of circulating glycine. This reduction is also observed in nondiabetic children of parents diagnosed with T2DM. A similar pattern of diminished plasma glycine concentrations is present in cases of obesity, where overweight and obese individuals have lower circulating glycine levels compared to their lean counterparts. Moreover, longitudinal studies conducted over periods ranging from 1 to 19 years reveal that lower glycine levels are inversely related to insulin resistance and serve as predictors for the development of T2DM [32].

#### 4.1.2. Serum TRP and Serum KA as Possible Biomarkers Indicating Renal Endothelial Injury in the Initial Stages of DKD

The biological implications of the kynurenine pathway activation in the context of progressive DKD are still not fully understood. Tryptophan undergoes metabolism through the kynurenine pathway, resulting in the formation of oxidized NAD. Since a reduction in NAD levels significantly contributes to the development of renal diseases, it is likely that many metabolites produced in the kynurenine pathway are redirected toward the NAD pathway to restore cellular NAD levels. Interestingly, manipulating indoleamine 2,3-dioxygenase (IDO) activity has led to conflicting results in various animal studies. For instance, the inhibition of IDO using 1-methyltryptophan exacerbated crescentic glomerulonephritis in renal tissues, which implies that the activation of IDO may exert an anti-inflammatory effect. Conversely, both the overexpression of IDO and the administration of IDO agonists appeared to alleviate renal damage in mice with IgA nephropathy. Additionally, a different preclinical investigation indicated that IDO activation could stimulate Wnt/β-catenin signalling, thereby promoting kidney fibrosis. While we cannot fully reconcile these opposing results due to lack of data deriving from human studies, the variability in the animal models utilized in these studies may help explain these inconsistencies [33].

### 4.2. Biomarkers Indicating Podocyte Injury and Proximal Tubule Dysfunction Evaluated Through Urine Analysis

#### 4.2.1. The Role of Urinary Glycine in Podocyte Injury and Proximal Tubule Dysfunction

Decreased glycine may be a secondary consequence of excessive free fatty acid (FFA) and branched-chain amino acids (BCAAs) metabolism. FFA metabolism can lead to an accumulation of *β*-oxidation intermediates (such as acyl-CoA esters). These are conjugated with glycine via the activity of acyl-CoA/glycine-*N*-acyltransferase, which is responsible for the transesterification of acyl-CoA esters with glycine to produce acyl-glycines. Acyl-glycine is membrane permeable and is readily excreted in the urine, thus serving as a mechanism for the elimination of excess *β*-oxidation intermediates. Similarly, BCAA catabolism byproducts also conjugate to glycine in the liver for excretion. Therefore, circulating glycine could be consumed to facilitate excretion of waste products associated with elevated FFAs and BCAAs. Indeed, in a rat model of obesity, a BCAA-restricted diet causes an increase in circulating glycine concentrations, demonstrating an inverse relationship between BCAA availability and circulating glycine levels. The BCAA-restricted diet restores acyl-glycines in the urine and the urinary acyl-glycine levels are linearly related to skeletal muscle glycine concentrations. These pathways have recently been reviewed. Regardless of the mechanism(s), it is important to note that the low circulating glycine levels in prediabetes/T2D are reflected in peripheral tissues and, to a lesser extent, in the brain, where key actions may be exerted on blood glucose control [34].

#### 4.2.2. Urinary Tiglylglycine

Tiglylglycine is a metabolite that appears in urine under certain pathological conditions. Its production is associated with metabolic disorders involving BCAA metabolism, particularly isoleucine catabolism. Isoleucine is a branched-chain amino acid metabolized in the mitochondria through several enzymatic steps. During its breakdown, isoleucine is converted to tiglyl-CoA, a critical intermediate. Normally, tiglyl-CoA is further metabolized by the enzyme enoyl-CoA hydratase in the β-oxidation pathway. If there is a defect in this pathway (e.g., a deficiency in enzymes like methylcrotonyl-CoA carboxylase or propionyl-CoA carboxylase), tiglyl-CoA accumulates. The accumulated tiglyl-CoA is detoxified through conjugation with glycine, forming tiglylglycine, which is then excreted in urine [35].

However, a case report describes an adult with beta-ketothiolase deficiency who developed diabetic ketoacidosis. This case highlights the importance of considering inherited metabolic disorders in the differential diagnosis of diabetic ketoacidosis, although such situations are rare. In conclusion, although tiglylglycine is not commonly associated with diabetes, monitoring its levels may be relevant in identifying rare metabolic disorders that may coexist or influence diabetes management [36].

### 4.3. A Concise Summary of the Clinical Significance of Metabolites

As it was mentioned by Imenshahidi, M., Gannon MC, and González-Ortiz M., glycine supplementation has been shown to enhance several aspects of metabolic syndrome, such as diabetes, obesity, hyperlipidemia, and hypertension by increasing insulin secretion. Looking ahead, glycine could play a crucial role in the clinical management of individuals suffering from metabolic syndrome [37,38,39].

A study conducted by Tomoko Inubushi et al. indicated that the administration of L-TRP to diabetic mice reduces the increase in glucose levels in blood and alleviates the strain on insulin secretion from beta cells [40].

To reduce misinterpretation of these substances and evaluate the clinical role, it is essential to conduct prospective randomized controlled trials.

The research presents a number of limitations: first it is a cross-sectional study, which means it cannot determine a causal relationship between our results and the clinical parameters. Second, the variability of the biological parameters (glycemia, lipids), could have introduced a bias in the interpretation of data. Third, the number of patients was limited.

Nevertheless, our study possesses certain strengths. To our knowledge, this is the first study that identifies tiglylglycine as a possible biomarker for DKD. Additionally, through statistical analysis, we demonstrated that this metabolite holds significance in effectively differentiating between various patient subgroups and exhibits robust correlations with markers related to proximal tubules and podocytes.

## 5. Conclusions

In this study, we utilized LC-UHPLC techniques to analyse the trend in serum and urine metabolites, aiming to identify the distinctions between control groups and various patient subgroups with diabetic kidney disease, with a specific focus on the normoalbuminuric subgroup. Our findings indicated several metabolites that may serve as putative biomarkers for early DKD, including glycine, kynurenic acid, tryptophan, and tiglylglycine.

## Figures and Tables

**Table 1 biomedicines-13-00675-t001:** Demographic data, clinical parameters, and biological results of healthy controls and of patients with type 2 DM divided into three stages: normoalbuminuria (P1), microalbuminuria (P2), and macroalbuminuria (P3).

	Healthy Subjects (C = 20)	Normoalbuminuria (P1 = 30)	Microalbuminuria (P2 = 30)	Macroalbuminuria (P3 = 30)
		clinical parameters		
Age (years)	68.0 ± 3.9	68.5 ± 4.9	69.6 ± 5.0	69.4 ± 3.9
Sex (%) M F	14 (70.0%) 6 (30.0%)	16 (53.3%) 14 (46.7%)	17 (56.7%) 13 (43.3%)	21 (70.0%) 9 (30.0%)
Duration of DM (years)	0 (0–0) # ▲	12 (10–17)	16 (14–22) ‡	22 (18–26)
BMI (kg/m^2^)	24.8 ± 4.4 # ▲	30.0 ± 4.5	31.5 ± 4.0	31.0 ± 5.3
Retinopathy (%)	0 (0.0%) ◊ ▲	5 (16.7%)	10 (33.3%) ¶	24 (80.0%)
Neuropathy (%)	0 (0.0%) # ▲	13 (43.3%)	15 (50.0%)	16 (53.3%)
		biological parameters		
Cholesterol (mg/dL)	132.5 ± 24.6 ▲	164.1 ± 53.5	166.8 ± 57.7 ‡	199.5 ± 48.1
HDLc (mg/dL)	45 (35–48)	46 (41–52)	44 (34–52)	45 (36–52)
LDLc (mg/dL)	68 (65–88) ∆	87 (65–114)	95 (62–120)	97 (70–137)
Triglycerides	98 (92–101) ◊ ▲	142 (105–172)	146 (112–203) ‡	188 (160–281)
HbA1c (%)	5.0 ± 0.2 # ▲	7.2 ± 0.9 *****	8.2 ± 1.4	8.4 ± 1.1
eGFR (mL/min/1.73 m^2^)	84 (81–88) # ▲	78 (73–83) **□**	67 (64–73) ¶	48 (42–56)
UA	11 (9–14) ◊ ▲	15 (11–22) **□**	69 (51–167) ¶	483 (263–1572)
UCr	93 (77–131)	89 (53–116)	100 (66–115)	70 (52–90)
uACR (mg/g)	11 (10–14) # ▲	20 (13–26) □	98 (61–155) ¶	925 (416–1391)
		markers of proximal tubular dysfunction		
KIM-1 (pg/g)	43 (26–46) # ▲	80 (67–94) □	140 (125–152) ¶	408 (321–458)
NAG (ng/g)	3 (2–3) ◊ ▲	3 (2–5) □	11 (11–18) ‡	17 (17–18)
		markers of endothelial damage		
U P-selectin	0 (0–0) # ▲	1 (1–1) □	3 (3–4) ¶	7 (5–7)
S P-selectin	1 (0–1) # ▲	1 (1–1) □	4 (4–5) ¶	8 (7–9)
U VCAM1 (pg/g)	2 (2–3) # ▲	5 (4–9) □	11 (9–12) ¶	16 (13–18)
S VCAM1 (pg/g)	10 (9–11) ◊ ▲	11 (11–12) □	15 (14–16) ¶	21 (19–24)
		markers of podocyte damage		
Podocalixin (mg/g)	0 (0–0) **#** ▲	1 (1–1) □	4 (3–4) ¶	8 (7–9)
Nephrin (ng/g)	0 (0–0) ▲	0 (0–0) □	0 (0–0) ¶	0 (0–0)
		METABOLITES		
s Glycine	217 (200–229) ▲	199 (169–204)	185 (169–199)	184 (170–187)
s Taurine	87 (83–93)	85 (82–90)	87 (76–91)	83 (78–87)
s KA	6 (4–6) ◊ ▲	4 (3–6)	4 (3–5)	4 (3–5)
s AC	5 (4–7)	6 (4–7)	5 (4–6)	5 (4–6)
s TRP	57 (48–67) ◊	51 (41–59)	52 (33–62)	48 (37–62)
u Glycine	12 (9–14) ∆	10 (7–14)	8 (6–12)	7 (5–13)
u Tiglylglycine	5 (4–6) ▲	4 (3–6)	4 (3–5) ¶	2 (1–3)
u Taurine	8 (6–10)	9 (7–10)	8 (6–11)	9 (8–14)
u KA	0 (0–0) ∆	0 (0–1)	0 (0–0)	0 (0–1)
u AC	0 (0–0)	0 (0–0)	0 (0–0)	0 (0–1)
u TRP	7 (4–10) ◊ ▲	4 (3–6) □	12 (6–16)	16 (10–21)

Continuous data presented as means ± SD or as median (interquartile range) as appropriate. *p*-value based on one-way ANOVA with Bonferroni correction, chi-squared test, Kruskal–Wallis test; significance of C vs. P1: # *p* < 0.001; ◊ *p* > 0.001 and *p* < 0.05; significance of P1 vs. P2: □ *p* < 0.001; ***** *p* > 0.001 and *p* < 0.05; significance of P2 vs. P3: ¶ *p* < 0.001; **‡** *p* > 0.001 and *p* < 0.05; significance of C vs. P1 vs. P2 vs. P3: ▲ *p* < 0.001; ∆ *p* > 0.001 and *p*< 0.05; BMI: body mass index; DM: diabetes mellitus; HbA1C: glycated hemoglobin; eGFR: estimated glomerular filtration rate; UA: urinary albumin; UCr: urinary creatinine; uACR: urinary albumin/creatinine ratio; KIM-1: kidney injury molecule-1; NAG: N-acetyl-β-(D)-glucosaminidase; U P-selectin: urinary p-selectin; S P-selectin: serum p-selectin; U VCAM-1: urinary vascular cell adhesion molecule-1; S VCAM-1: serum vascular cell adhesion molecule-1; s Glycine: serum glycine; s Taurine: serum taurine; sKA: serum kinurenic acid, sAC: serum acetylcarnitine; sTRP: serum tryptophan; u glycine: urinary glycine; u tiglyglycine: urinary tiglylglycine; u taurine: urinary taurine; u KA: urinary kinurenic acid; u AC: urinary acetylcarnitine; u TRP: urinary tryptophan.

**Table 2 biomedicines-13-00675-t002:** Univariable linear regression analysis of serum and urine samples.

Dependent Variable	Independent Variable	R^2^	Coef β	*p* Value
s Glyc	p-selectin	0.067	−3.14	0.006
	VCAM1	0.069	−2.01	0.005
s Taurine	p-selectin	0.020	−0.80	0.138
	VCAM1	0.010	−0.35	0.296
s KA	p-selectin	0.099	−0.12	0.000
	VCAM1	0.061	−0.06	0.008
s AC	p-selectin	0.000	−0.005	0.928
	VCAM1	0.003	−0.02	0.553
s TRP	p-selectin	0.003	−0.29	0.548
	VCAM1	0.007	−0.27	0.372
u Glyc	Nephrin	0.039	−0.68	0.037
	Podocalixin	0.025	−0.36	0.097
	KIM1	0.054	−0.01	0.014
	NAG	0.027	−0.16	0.081
u Tiglyglycine	Nephrin	0.156	−0.57	0.000
	Podocalixin	0.120	−0.33	0.000
	KIM1	0.178	−0.00	0.000
	NAG	0.051	−0.09	0.017
u Taurine	Nephrin	0.005	0.19	0.428
	Podocalixin	0.031	0.30	0.065
	KIM1	0.040	0.00	0.034
	NAG	0.032	0.13	0.059
u KA	Nephrin	0.001	0.00	0.681
	Podocalixin	0.032	0.01	0.057
	KIM1	0.023	0.00	0.111
	NAG	0.027	0.00	0.084
u AC	Nephrin	0.039	0.03	0.036
	Podocalixin	0.052	0.02	0.015
	KIM1	0.036	0.01	0.045
	NAG	0.098	0.01	0.000
u TRP	Nephrin	0.019	0.72	0.148
	Podocalixin	0.044	0.72	0.027
	KIM1	0.052	0.01	0.015
	NAG	0.035	0.28	0.047

KIM-1: kidney injury molecule-1; NAG: N-acetyl-β-(D)-glucosaminidase; P-selectin: p-selectin; VCAM-1: vascular cell adhesion molecule-1; s Glycine: serum glycine; s Taurine: serum taurine; sKA: serum kinurenic acid, sAC: serum acetylcarnitine; sTRP: serum tryptophan; u glycine: urinary glycine; u tiglyglycine: urinary tiglylglycine; u taurine: urinary taurine; u KA: urinary kinurenic acid; u AC: urinary acetylcarnitine; u TRP: urinary tryptophan.

**Table 3 biomedicines-13-00675-t003:** Urinary tiglylglycine; p-selectin, podocalixin, kidney injury molecule-1, significance of urinary tiglylglycine, and correlation with p-selectin, podocalixin, KIM-1 (*p* < 0.01).

	Dependent Variable	R^2^	Coef β	*p* Value
U Tiglylglycine	P-selectin	0.2143	−1.49	0.0000
	Podocalixin	0.2143	1.47	0.0000
	KIM1	0.2143	−0.01	0.0000

## Data Availability

The data that support the findings of this study are available from the corresponding author upon reasonable request.

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
