# Peer review of "Urinary and Serum Amino Acids May Be Associated with Podocyte, Proximal Tubule, and Renal Endothelial Injury in Early Diabetic Kidney Disease in Type 2 Diabetes Mellitus Patients"

_biomedicines, 2025, doi:10.3390/biomedicines13030675_

Round 1
Reviewer 1 Report
Comments and Suggestions for Authors
With the increasing incidence of Diabetes Mellitus World wide, Diabetes Kidney Disease (DKD) has emerged as a major health threat. As its the leading causing of chronic kidney disease there is a need for better diagnostic markers for DKD that can help in early diagnosis and determine the course of treatment. In this study the authors have successfully utilized Mass spectrometry and ELISA to identify novel metabolites that have the potential to serve as biomarkers. This study is especially significant as the authors have further draw potential links between these identified metabolites and associated podocytes, endothelial and tubule dysfunction. While the manuscript is ready for publication , I have the following concerns/suggestions that if addressed could improve the quality of the manuscript for larger audiences -
- In Table 1 that lists metabolites , the authors should include a "fold-change" graph compared to healthy individuals which would indicate the exact range of these metabolites like KIM-1 etc. that would be associated with dysfunction and that clinicians can prefer back to.
- The authors fail to mention about the exact treatment dosage /time period etc ( only mention- angiotensin )received by these patients that can help the readers draw links between treatment options and DKD.
- The p value of 0 seems erroneous in Table 3 and needs to be double checked.
- What is the advantage between using these new metabolite markers vs the already available options like creatinine etc ?
- The authors need to clarify if KIM-1 and NAG were found both in the serum and urine ? Its not clear to me.
- Minor - The full form of T2DM should be mentioned at least once in the beginning.
Author Response
Thank you for your questions!
- In Table 1 that lists metabolites, the authors should include a "fold-change" graph compared to healthy individuals which would indicate the exact range of these metabolites like KIM-1 etc. that would be associated with dysfunction and that clinicians can prefer back to.
We cannot include a fold change graph in the study because markers of endothelial dysfunction, podocyte and tubular damage (VCAM1, p-selectin, podocalixin, nephrin, KIM-1, NAG) are presented as medians and interquartile ranges (IQR) for variables with a skewed distribution.
- The authors fail to mention about the exact treatment dosage /time period etc (only mention- angiotensin) received by these patients that can help the readers draw links between treatment options and DKD.
The duration of therapy involving ACE inhibitors (eg.: Perindopril) or sartans (eg.: Candesartan, Irbesartan) should be no less than 10 years, with dosages tailored according to the individual blood pressure management of each patient.
- The p value of 0 seems erroneous in Table 3 and needs to be double checked.
I reviewed the p-value listed in the table, and the results from the statistical analysis confirm the value presented there.
- What is the advantage between using these new metabolite markers vs the already available options like creatinine etc ?
I aim to utilize these metabolites as an addition to creatine and cystatin C, since creatinine and cystatin C levels rise following kidney injury. These metabolites serve to indicate early alterations in kidney function, even among patients with normal albumin levels. This approach could aid in the prompt treatment and prevention of the progression or worsening of chronic kidney disease and diabetes.
- The authors need to clarify if KIM-1 and NAG were found both in the serum and urine ? Its not clear to me.
kidney injury molecule-1 (KIM-1), N-acetyl-β-(D)-glucosaminidase (NAG), were quantified in peripheral blood and urine by qRT-PCR.
- Minor - The full form of T2DM should be mentioned at least once in the beginning
I have modified in the article as recommended.

Reviewer 2 Report
Comments and Suggestions for Authors
The study by Mogos et al. presents valuable insights in the field of biomarkers. The authors identify tiglylglycine, along with other molecules such as glycine, kynurenic acid, and tryptophan, as potential biomarkers for early diabetic kidney disease (DKD). Furthermore, they demonstrate that the metabolite tiglylglycine shows strong correlations with markers associated with proximal tubules and podocytes. This metabolite could also be useful for distinguishing between different patient subgroups.
The study is well written and identifying new biomarkers for diabetes-related diseases represents a big challenge. I only have a few minor comments on the study.
- I recommend presenting some of the relevant data in graphs rather than in a table to make it easier and more immediate for the reader to understand. Alternatively, a graphical abstract could also be helpful.
- Please correct the typo on line 156: "p-selectina."
- Line 220 AK=KA?
Author Response
Thank you for your suggestions!
- I recommend presenting some of the relevant data in graphs rather than in a table to make it easier and more immediate for the reader to understand. Alternatively, a graphical abstract could also be helpful.
The graphical abstract is attached in the beggining of the article.
- Please correct the typo on line 156: "p-selectina."
I modified in the article as recommended.
- Line 220 AK=KA?
Yes. It was a minor mistake in writing the article.

Reviewer 3 Report
Comments and Suggestions for Authors
Introduction: More citations are required to smoothly forward the readers to other sections of the paper.
Methodology:
Overall, authors have included 110 participants in this study, from which 20 were healthy control subjects and 90 were patients with long-standing T2DM. "Based on UACR, we stratified the 90 patients with T2DM in 3 equal subgroups of 30 patients each: subgroup P1 (UACR<30 mg/g), P2 (UACR: 30–300 mg/g) and P3 (UACR > 300 mg/g). As an inclusion criterion, we selected patients with T2DM, with a duration of minimum five years. In addition, hospitalized patients enrolled in the study were non-critically ill, and the samples were collected after initial biological investigations. Patients with T2DM with poor control of diabetes (HbA1c > 10%), active infections, neoplasia, glomerular disease, end-stage renal disease, and T1DM were excluded from the study".
The rationale behind choosing this sample size should be explained. How can authors be sure that this study has enough power? Considering the fact that you have included patients with specific inclusion criteria, please discuss the limitations and generalizability of your study.
Very minor issues: you may use more commonly used symbols to indicate significances and p values instead of ● ♣.
Correct the title of Table 3. u tiglyglycine: urinary tiglylglycine; ; P-selectin: p-selectin, KIM-1: kidney injury molecule1.
Author Response
Thank you for your suggestions!
- The rationale behind choosing this sample size should be explained. How can authors be sure that this study has enough power? Considering the fact that you have included patients with specific inclusion criteria, please discuss the limitations and generalizability of your study.
We screened 170 patients to verify the eligibility of the participants. From their total, 90 patients were selected who met the criteria we established. The research presents a number of limitations: first it is a cross-sectional study, which means it cannot determine a causal relationship between our results and the clinical parameters. Second, the variability of the biological parameters (glycemia, lipids,) could have introduced a bias in the interpretation of data. Third, the number of patients was limited (data shown in the subsection 4.3.)
- Very minor issues: you may use more commonly used symbols to indicate significances and p values instead of ● ♣.
I modified in the article as recommended.
- Correct the title of Table 3. u tiglyglycine: urinary tiglylglycine; ; P-selectin: p-selectin, KIM-1: kidney injury molecule1.
I corrected in the text as suggested.
